# Dietary Methionine Deficiency Enhances Genetic Instability in Murine Immune Cells

**DOI:** 10.3390/ijms22052378

**Published:** 2021-02-27

**Authors:** Regina L. Binz, Ratan Sadhukhan, Isabelle R. Miousse, Sarita Garg, Igor Koturbash, Daohong Zhou, Martin Hauer-Jensen, Rupak Pathak

**Affiliations:** 1Division of Radiation Health, Department of Pharmaceutical Sciences, College of Pharmacy, University of Arkansas for Medical Sciences, Little Rock, AR 72205, USA; rkbinz@uams.edu (R.L.B.); RSadhukhan@uams.edu (R.S.); GargSarita@uams.edu (S.G.); mhjensen@uams.edu (M.H.-J.); 2Department of Biochemistry and Molecular Biology, College of Medicine, University of Arkansas for Medical Sciences, Little Rock, AR 72205, USA; iracinemiousse@uams.edu; 3Department of Environmental and Occupational Health, College of Public Health, University of Arkansas for Medical Sciences, Little Rock, AR 72205, USA; ikoturbash@uams.edu; 4Center for Dietary Supplements Research, University of Arkansas for Medical Sciences, Little Rock, AR 72205, USA; 5Department of Pharmacodynamics, College of Pharmacy, University of Florida, Gainesville, FL 32610, USA; zhoudaohong@cop.ufl.edu

**Keywords:** monocytes, macrophages, radiation, chromosomal instability, DNA damage and repair, spectral karyotyping, G-banding

## Abstract

Both cell and animal studies have shown that complete or partial deficiency of methionine inhibits tumor growth. Consequently, the potential implementation of this nutritional intervention has recently been of great interest for the treatment of cancer patients. Unfortunately, diet alteration can also affect healthy immune cells such as monocytes/macrophages and their precursor cells in bone marrow. As around half of cancer patients are treated with radiotherapy, the potential deleterious effect of dietary methionine deficiency on immune cells prior to and/or following irradiation needs to be evaluated. Therefore, we examined whether modulation of methionine content alters genetic stability in the murine RAW 264.7 monocyte/macrophage cell line in vitro by chromosomal analysis after 1-month culture in a methionine-deficient or supplemented medium. We also analyzed chromosomal aberrations in the bone marrow cells of CBA/J mice fed with methionine-deficient or supplemented diet for 2 months. While all RAW 264.7 cells revealed a complex translocation involving three chromosomes, three different clones based on the banding pattern of chromosome 9 were identified. Methionine deficiency altered the ratio of the three clones and increased chromosomal aberrations and DNA damage in RAW 264.7. Methionine deficiency also increased radiation-induced chromosomal aberration and DNA damage in RAW 264.7 cells. Furthermore, mice maintained on a methionine-deficient diet showed more chromosomal aberrations in bone marrow cells than those given methionine-adequate or supplemented diets. These findings suggest that caution is warranted for clinical implementation of methionine-deficient diet concurrent with conventional cancer therapy.

## 1. Introduction

Methionine is an essential sulfur-containing amino acid that regulates a number of key cellular functions, including, but not limited to, protein synthesis (as the first amino acid of all newly synthesized polypeptide chains), methylation of both DNA and RNA, and generation of polyamines [1]. Because methionine regulates a variety of key cellular functions, tumor control with complete or partial deficiency of dietary methionine has been a major focus of research for decades.

While a large number of in vitro and preclinical studies demonstrate that dietary methionine deficiency inhibits tumor growth [2,3,4], the clinical success of this strategy is not well-established. Moreover, the risks associated with this dietary intervention on healthy cells, specifically those involved in modulating tumor growth, are unknown. 

Monocytes are a type of innate immune cells that originate from hematopoietic stem and progenitor cells of bone marrow. Under normal physiological conditions, monocytes circulate in the blood for around 1 day and, during this time, some of the circulating monocytes infiltrate into various tissues for immune surveillance, where they subsequently differentiate into macrophages or microglia when they present in the brain. 

During tumorigenesis, monocytes infiltrate into tumors [5,6]. These monocytes then transform into tumor-associated macrophages. Increased macrophage prevalence correlates with poor overall survival in many human cancers. Notably, tumor-associated macrophages, depending on their types, can exert either pro-tumor or anti-tumor activity by altering the expression of various proteins [5,6]. Protein expression is tightly regulated by chromosomal DNA. Genetic and epigenetic changes in DNA, such as changes in DNA methylation, can alter protein expression [7,8]. By exploiting the relationship between nutrition and gene expression, the epigenetic phenomena related to growth and health processes can be regulated and this dynamic relationship between nutrition and gene expression throughout the lifetime of an organism has now been recognized as a subfield called “nutritional epigenomics” [9]. The enigma is that both DNA hypomethylation and hypermethylation have been shown to have negative effects on chromosome structure and protein expression [10]. Any change in DNA methylation status may negatively affect the expression of proteins that maintain the structural integrity of chromosomes. DNA hypomethylation is associated with numerical and structural chromosomal aberrations [10], which are considered crucial markers for genetic instability. Multiple studies have shown that deficiency of dietary methionine leads to hypomethylation of mitochondrial and nuclear DNA [11,12]. On the contrary, dietary methionine supplementation may result in either DNA hypo- or hyper-methylation [13], which also causes chromosomal aberrations. Abnormal synthesis of proteins involved in chromosome segregation during anaphase may lead to loss or gain of entire chromosome(s), resulting in aneuploidy. However, the effects of dietary methionine deficiency on genetic stability in immune cells remain unknown. 

The assessment of chromosomal integrity provides insight into the genotoxic effects of various external or internal stimuli. Chromosomes are prepared by arresting proliferating cells during metaphase using a tubulin inhibitor to prevent spindle fiber formation. During metaphase, chromosomes are condensed such that most structural or numerical abnormalities can be identified. Such aberrations include deletions, duplications, and rearrangements (translocation of one chromosome segment to the same or another chromosome). Rearrangements can present in the form of derivative (Der) chromosomes, which are generated as a result of more than one structural aberration within a single chromosome or an unbalanced translocation (T) involving two or more chromosomes [14]. These aberrations are best identified by using various methods of analysis, as each technique has limitations. Solid Giemsa staining provides a uniform purple color to each chromosome, while trypsin–Giemsa banding adds the advantage of revealing the unique pattern of horizontal light and dark bands along the length of each individual chromosome. Any deviation from the normal banding pattern is considered a rearrangement. Depending on the band level of resolution, G-banding has limitations in differentiating both interchromosomal aberrations (within different chromosomes) and intrachromosomal (within the same chromosome) aberrations. Molecular cytogenetic techniques are best used after Giemsa-banded chromosome analysis. Interchromosomal rearrangements can be best detected by a molecular cytogenetic technique called spectral karyotyping (SKY). This technique allows simultaneous hybridization of all chromosomes, using various combinations of five different fluorochromes, to target chromosome-specific chromatin, resulting in unique spectral emission for each chromosome [14]. The spectral emission is measured, and a distinct pseudocolor is assigned to each chromosome. While effective in determining interchromosomal rearrangements, SKY is limited by the band level of resolution, and results should be further elucidated by fluorescent in situ hybridization (FISH). FISH utilizes the hybridization of locus-specific DNA probes tagged with a fluorescent marker to hybridize to a specific DNA sequence. While this technique is useful for both interphase and metaphase analysis, metaphase FISH is more reliable than interphase FISH. Radiation plays an integral role in cancer therapy. Unfortunately, it is a genotoxic agent which generates a variety of reactive oxygen species (ROS) within milliseconds of exposure. This causes DNA damage, with double-strand breaks (DSBs) recognized as the most detrimental [15]. During radiotherapy, some toxicity to normal cells, including monocytes/macrophages, is inevitable. DSBs are immediately recognized by cellular DNA repair machinery [16]. Effective repair of DSBs results in cell survival, while inadequate repair leads to chromosomal aberrations or cell death. DSBs are repaired either through homologous recombination (HR) or through non-homologous end joining (NHEJ)—the choice of which is determined by the cell cycle phase and chromatin context. NHEJ is the preferred mechanism for the repair of DSBs in the G1 phase of the cell cycle when a DNA template that could be used for HR is absent [17]. Importantly, the effects of dietary methionine deficiency or supplementation in radiation-induced DSB formation and chromosomal aberration of immune cells have not been examined. 

In the current study, we provide a cytogenetic characterization of the mouse RAW 264.7 monocyte/macrophage cell line. We used G-banding for identification/quantification of numerous chromosomal aberrations in RAW 264.7 cells after modulating methionine content in the culture medium followed by either sham irradiation or single exposure. SKY analysis was utilized for further confirmation of G-banding results. Molecular studies were performed to determine the effects of dietary methionine modulation in DNA damage and repair with or without irradiation. Finally, chromosomal damage in the bone marrow cells of mice fed with either a normal diet (methionine-adequate diet), methionine-deficient, or methionine-supplemented diet for two months were assessed with solid Giemsa staining. These findings will augment our knowledge of the RAW 264.7 karyotype, improve our understanding regarding the effects of dietary methionine modulation on chromosomal integrity, and provide critical information for considering the potential risk of dietary methionine deficiency during cancer treatment.

## 2. Results

### 2.1. RAW 264 Cell Line Exhibits a Consistent Three-Way Translocation

First, we characterized the mouse RAW 264.7 monocyte/macrophage cell line by conventional G-banding. Results were validated using SKY. G-banding permits the identification of a change in the banding pattern, indicating chromosomal rearrangements/translocations. We observed a consistent complex aberration (involving ≥ three breaks in ≥ two chromosomes) involving chromosome 1, 3, and 8 after scoring 35 metaphase spreads. Precisely, we observed a distal deletion at one of chromosome 1, an addition to one homologue of chromosome 3, as well as a large additional segment at the distal end of one chromosome 8 homologue, as evident by changes in banding pattern compared to the homologue of each chromosome (Figure 1A). Because G-banding cannot identify interchromosomal rearrangements accurately, we used SKY to confirm G-band analysis. SKY precisely revealed the chromosomes involved in the consistent complex aberration in RAW 264.7 cells. First, the distal segment of one of chromosome 1 homologue has translocated to the distal end of one copy of chromosome 3 homologue, thus forming a distal end deleted chromosomal 1 and a derivative chromosome 3 (i.e., Der(3)T(1;3)) (Figure 1B,C). Finally, the additional material on one homologue of chromosome 8 was shown to be consistent in spectral emission to chromosome 3 chromatin, forming a Der(8)T(3;8) (Figure 1B,C). Notably, the translocated piece of chromosome 3 on chromosome 8 is much larger in size than the original chromosome (Figure 1B).

### 2.2. RAW 264 Cell Line Harbors Three Different Clones Based on Chromosome 9

Careful observation of 35 karyograms utilizing G-band analysis revealed that the RAW 264.7 cell line can be divided into three different clones based on morphology/banding pattern of chromosome 9. Clone 1 represents no structural change in homologous pairs of chromosome 9 (Figure 2). Clone 2 reveals that one copy of chromosome 9 is much longer than the other homologue, with a distinct, dark-stained constriction toward the distal end (Figure 2). Clone 3 also shows a larger copy of chromosome 9, with a unique banding pattern, but no distinct constriction (Figure 2). SKY analysis revealed the unknown banding pattern observed for chromosome 9 in both clones 2 and 3 to be consistent with spectral emission of chromosome 9, thus concluding that the aberrant banding pattern does not originate from another chromosome (Figure 2). Notably, the distribution of clones 1, 2, and 3 was 77% (27/35), 20% (7/35), and 3% (1/35), respectively, when RAW 264 cells were grown in media with a normal level of methionine (Appendix A).

### 2.3. Methionine Deficiency Increases Structural Chromosomal Aberrations and Alters the Ratio of Three Clones

To determine the effects of methionine on chromosomal aberration, we cultured RAW 264.7 cells for 1 month either in methionine-deficient (0.25×) or in methionine-adequate (1×) media. Because modulation of dietary methionine content alters DNA methylation, a critical determinant for maintaining chromosomal integrity, we predicted that dietary intervention of methionine would impair the structural integrity of chromosomes. We observed an increase in the frequency of chromosomal aberrations in RAW 264.7 cells following 1-month culture in 0.25× media as detected by G-band analysis. Methionine-deficient (0.25×) media resulted in five Robertsonian translocations (Rb) and two breaks on chromosome 9 out of 36 cells analyzed, while only 1 Rb and one gain of chromosome 19 were observed out of 35 cells analyzed when RAW 264.7 cells were cultured in methionine-adequate (1x) media (Table 1). Because DSBs serve as one of the main drivers for chromosomal aberrations, we determined whether methionine deficiency enhances DSB formation in RAW 264.7 cells. We found that RAW 264.7 cells had increased numbers of DSBs after 1-month culture in 0.25× medium based on the analysis of the protein level of phosphorylated gamma-H2AX with Western blot (Figure 3). Additionally, we checked the effects of 3× media on DSB formation. We found that 1-month culture of RAW 264.7 cells in 3x media suppressed DSB formation (Figure 3). Finally, we observed that 1-month culture of RAW 264.7 cells in 0.25× media alters the ratio of three clones as compared to cells grown in 1× media. The distribution of clones 1, 2, and 3 was 42% (15/36), 50% (18/36), and 8% (3/36), respectively, when RAW 264 cells were grown in 0.25x media for 1 month. (Appendix A).

### 2.4. Methionine Deficiency Enhances Radiation-Induced Chromosomal Aberrations and DSB Formation

Radiation induces DSBs and chromosomal aberrations. Because methionine deficiency increased chromosomal aberrations and DSBs, we assumed that radiation would further increase the frequency of these two parameters. Indeed, methionine deficiency resulted in an increase in radiation-induced chromosomal aberrations and DSBs. One-month culture of RAW 264.7 cells in 0.25× media followed by exposure to 2 Gy radiation resulted in 1.32 acentric fragment formation per cell (50/38), while the same radiation dose resulted in only 0.83 acentric fragment/cell (30/36) when cultured in 1.00× media (Table 2; Appendix A). Radiation also increases the number of color junctions in RAW 264.7 cells when grown in 0.25× media, as compared to 1× media (Table 2; Appendix A). Radiation also elevates DSB formation in RAW 264.7 cells (as detected by Western blot of phosphorylated gamma-H2AX protein) to a greater extent when cultured in 0.25× media compared to 1× media (Figure 4A,B). Immunocytochemistry for phosphorylated gamma-H2AX further confirmed our Western blot data (Figure 4C). Because DSBs are immediately recognized by the MRN complex, we predicted that the modulation of methionine in the media would alter the expression of MRN complex members following irradiation. After one month of culture of RAW 264.7 cells in 0.25× media, we observed a significant increase in MRE11 expression as compared with cells grown in either 1.00× or 3.00× media (Figure 4D,E).

### 2.5. Dietary Methionine Deficiency Elevates Chromosomal Damage in the Bone Marrow Cells of Mice

Finally, we extended our study in vivo. We fed three groups of male CBA/J mice with methionine-deficient (MDD), adequate (MAD), and supplemented (MSD) diet for 2 months, then collected bone marrow cells from the mice for chromosome studies. We observed that dietary methionine deficiency caused around four-times more chromosomal aberrations versus those fed with methionine-adequate or supplemented diet (Table 3).

## 3. Materials and Methods

### 3.1. Cell Culture and Medium

RAW 264.7 cells were procured from ATCC (Manassas, VA, USA) and cultured in a specially formulated DMEM (Gibco, Gaithersburg, MD, USA; Cat# 21013024) with no glutamine, methionine, and cysteine. Three types of media based on methionine (Acros Organics, Lenexa, KS, USA; Cat# 166160250) content were prepared by adding either 0.05 mM methionine (methionine-deficient or 0.25× media) or 0.2 mM methionine (methionine-adequate or 1.00× media) or 0.6 mL methionine (methionine-supplemented or 3.00× media). Because complete removal of methionine stops RAW 264.7 cells’ growth, we added a very low concentration of methionine in 0.25× media. All 3 different types of media were supplemented with 0.2 mM l-cystine (Alfa Aesar™, Ward Hill, MA, USA; Cat# J62292), 4 mM glutamine (Gibco; Cat# 25030081), 1mM sodium pyruvate (Gibco; Cat# 11360070), and 10% dialyzed fetal bovine serum (Gibco; Cat# 26400044). Cells were maintained in a water-jacketed humidified incubator with 5% CO_2_ (Thermo Scientific, Memphis, TN, USA) at 37 °C. Cells were subcultured every 3 days after dissociation with a brief treatment with 0.25% trypsin solution (Gibco; Cat# 25200056). Experiments were performed between passage numbers 12 through 25.

### 3.2. Mice

Eight-week-old male CBA/J mice were purchased from Jackson Laboratory (Bar Harbor, ME, USA). Mice were initially housed in groups of 2 to 4 animals per cage. For the duration of the study, the animals had free access to food and water and were maintained under a 12:12 light:dark cycle. All procedures were approved by the Institutional Animal Care and Use Committee at UAMS. The animal protocol (Animal Use Protocol #3823) was approved by the Institutional Animal Care and Use Committee of the University of Arkansas for Medical Sciences.

### 3.3. Diet

Isocaloric methionine-adequate (MAD, TD.140520), methionine-deficient (MDD, TD.90262, 0% Met from normal), and methionine-supplemented (MSD, TD. 160241, 300% met from normal) diets were purchased from Envigo Teklad Diets (Madison, WI, USA) and were provided to animals ad libitum. Diet compositions are provided in Appendix A.

### 3.4. Irradiation

Cells were grown in non-tissue culture T25 flasks (Corning, NY, USA) and were exposed to 2 Gy ionizing radiation with a Shepherd Mark I ^137^Cs irradiator (model 25, J. L. Shepherd & Associates, San Fernando, CA, USA). Sham-irradiated (0 Gy) cells served as a control. Flasks were placed on a turntable rotating at 6 rpm to ensure uniform dose distribution. The average dose rate was 1.01 Gy/min and was corrected daily for decay. At least once a year, dosimetry of the irradiator is performed with gafchromic film and alanine tablets, analyzed by the National Institute of Standards and Technology (NIST), and with ion chambers calibrated yearly in a NIST-traceable laboratory (University of Wisconsin). Dosimetry is overseen by Dr. Narayanasamy, board-certified medical physicist at UAMS.

### 3.5. Chromosome Preparation

Metaphase chromosomes were prepared as described elsewhere [14]. Cells were treated with KaryoMAX colcemid (Gibco; Cat# 15212012) at a final concentration of 75 ng/mL for 20 min to arrest cells in metaphase. Cells then were washed with PBS without Ca^2+^ and Mg^2+^ (Gibco; Cat# 14190250), followed by trypsin treatment to dislodge adherent cells, and centrifuged at 1000 rpm at room temperature. The supernatant was removed, and the cell pellet was gently resuspended in prewarmed hypotonic solution (75 mM KCl; Gibco; Cat# 10575090). Cells then were incubated in a 37°C water bath for 15 min. After hypotonic treatment, cells were fixed by gently adding 500 μL of acetomethanol fixative (3:1, methanol: acetic acid) with gentle and thorough agitation. After centrifugation for 10 min at 1000 rpm, the cell pellet was gently resuspended in 2 mL of supernatant and fixative was added in a dropwise manner with constant agitation for a total of 4 mL. An additional 4 mL of fixative was added and the cells were allowed to rest at room temperature for 20 min. After two additional washes with fixative solution, cells were dropped onto pre-cleaned glass slides at 20–25 °C with 42–45% humidity to obtain optimum spreading of metaphase chromosomes.

### 3.6. G-Banding

Trypsin–Giemsa staining was used to prepare G-banded chromosomes [14]. Slides were baked overnight at 66°C and treated with 0.025% trypsin for 1 min, gently rinsed with Tyrode’s buffer (Sigma, St. Louis, MO, USA), and stained with Giemsa (Sigma) for 5 min. Karyotype integrity of RAW 264.7 cells was determined after G-banding, according to the Rules for Nomenclature of Mouse Chromosome Aberrations (http://www.informatics.jax.org/mgihome/nomen/anomalies.shtml (accessed on 3 February 2021)). For karyotyping, images were acquired with a Zeiss Imazer.Z2 microscope (White Plains, NY, USA) equipped with the GenASIs Case Data Manager System, version 7.2.2.40970 (ASI, Carlsbad, CA, USA). At least 40 well-spread, randomly selected metaphase spreads were scored.

### 3.7. Spectral Karyotyping (SKY)

We used the probe and concentrated antibody detection kits from Applied Spectral Imaging (ASI) according to the manufacturer’s protocol; the probe mixture and hybridization reagents were prepared as recommended. Briefly, slides were aged at room temperature overnight. Just before hybridization, slides were soaked in 2× SSC (Sigma) at room temperature for 5 min and dehydrated in 70%, 80%, and 100% ethanol for 2 min each. Slides were denatured in pre-warmed denature solution (70% formamide [Millipore, Temecula, CA, USA] in 2× SSC [Fisher Scientific]) at 71 °C for 1 min and immediately placed into a cold ethanol series (70%, 85%, 100%; 2 min each). SKY probes (ASI) were denatured in a water bath at 80–81 °C for 7 min. After denaturation, 10 μl of denatured probe mixture was applied to the target area of the slide, covered with a 22 × 22-mm coverslip, and sealed with rubber cement (Elmers Rubber Cement [Fisher Scientific]; Cat# 50-189-9750). Slides were placed in a humidified light-protected box and allowed to hybridize at 37 °C for 48 h. Hybridized slides then were washed with formamide wash solution (50% formamide [Fisher Scientific] in 2× SSC [Fisher Scientific]) pre-warmed to 45 °C three times, followed by two washes in 1× SSC for 5 min each. Then, 80 μL of blocking reagent (ASI) was applied to the target area and protected with a plastic coverslip. Slides were incubated for 30 min at 37 °C. Cy5 and Cy5.5 antibody staining reagent (ASI) was reconstituted with filtered 4× SSC. The coverslip was carefully removed, and 80 μL Cy5 antibody solution (ASI) was applied to the target area and protected with a clean plastic coverslip. Slides were incubated for 1 h at 37 °C. Slides were washed three times with pre-warmed 4× SSC containing 0.1% Tween-20 for 5 min each at 45 °C. Cy5.5 antibody solution (ASI) was applied to the target area and protected with a plastic coverslip. The slides were again incubated for 1 h at 37 °C. The 4× SSC/0.1% Tween wash was repeated. DAPI counterstain was immediately applied, and a clean glass coverslip was placed over the target area. Image acquisition for SKY was performed with an SD200 Spectracube (ASI) mounted on a Zeiss Imager.Z2 microscope with SKY View software. SKY images were captured under 63× magnification. DAPI images were captured, inverted, and enhanced to produce G-band-like patterns on the chromosomes. The simultaneous visualization of all mouse chromosomes in different colors was achieved by spectral imaging. Spectral imaging combines fluorescence microscopy, CCD-imaging, and Fourier spectroscopy to visualize simultaneously the entire spectrum at all image points.

### 3.8. Aberration Scoring Technique

We scored various types of numerical and structural aberrations with conventional G-banding analysis and SKY at 63× magnification, as described elsewhere [14]. At least 30 metaphase spreads were scored for each staining technique. Loss or gain of chromosomal material is considered deletion or addition, respectively. A chromosomal aberration was considered complex if it involved three or more breaks in two or more chromosomes. Deletions, insertions, and terminal and reciprocal translocations were counted as simple aberrations. A rearrangement was counted as simple when it consisted of a maximum of two breaks in two chromosomes. The number of color junctions per cell is a simple parameter representing the frequency of improper rejoining of chromosomes, and the number of excess painted fragments represents acentric fragments.

### 3.9. Western Blot Analysis

Cells grown on 6-well plates were washed with PBS and lysed with RIPA lysis buffer (Boston Bioproduct; Cat# BP-115) containing a cocktail of protease and phosphatase inhibitor (Thermo Scientific; Cat# 1861280) at 4 °C for 25 min to extract protein. Cell extracts were resolved by SDS/PAGE (Bio-Rad, Hercules, CA, USA; Cat# 5671094) and transferred to PVDF membranes (Cytiva Amersham, Fisher Scientific; Cat# 10600023). The resulting membranes were incubated with primary antibodies for 16 to 18 h at 4 °C, washed with 1× TBST (Boston Bioproducts, Inc., Ashland, MA, USA; Cat# IBB-580X), and incubated for 1 h with goat anti-rabbit secondary antibodies (Cell Signaling Technology, Danvers, MA, USA; Cat# 7074) at room temperature prior to revelation using chemiluminescence (Thermo Scientific; Cat# 34580). 

### 3.10. Immunocytochemistry

Cells were grown in chamber slides (Thermo Scientific; Nunc™ Lab-Tek™ II Chamber Slide™ System; Cat# 154453). After radiation exposure, cells were washed with cold PBS twice. Cells were then fixed with 4% paraformaldehyde (Boston Bioproducts, Inc.; Cat# BM-155) for 10 min at 4 °C and washed again with cold PBS. After washing, 0.25% Triton X-100 (Acros Organics, Fisher Scientific; Cat# 327371000) was added to permeabilize the cells for 10 min at 4 °C. Permeabilized cells were then blocked with 1% BSA/0.3M glycine in PBS-T (PBS  +  0.1% Tween 20) for 1 h at 4 °C. Then, cells were incubated with anti-p γH2AX (Novus Biologicals, Centennial, CO, USA; Cat# NB100-384) in PBS-T containing 1% BSA for overnight. The next day, cells were washed with PBS-T and AF488 tagged secondary antibody (Invitrogen, [Thermo Fisher Scientific, Memphis, TN, USA]; Cat# A-11008) in PBS-T was added for 1 h. Cells were then washed and mounted with DAPI-containing mounting medium (Vector Laboratories, Burlington, ON L7N 3J5, Canada; Cat# H-1200). Images were captured with a Zeiss fluorescence microscope.

### 3.11. Statistical Analysis

Statistical analysis was performed with online GraphPad Prism software (San Diego, CA, USA). Aberration frequency was calculated by dividing the number of aberrations observed by the total number of metaphase spreads scored. Standard errors for the frequencies were calculated by $\sqrt{a}$*/A*, where “*a*” represents the number under consideration and *A* is the total number of metaphase spreads analyzed, as described elsewhere. Differences in induction of various types of aberrations between two groups were calculated with an unpaired *t*-test. Differences in aberration induction between two groups were considered statistically significant when *p* was less than 0.05.

## 4. Discussion

Immortalized cell lines are important research tools for in vitro and in vivo studies. Generation of an immortalized cell line requires the genetic manipulation of normal cells, which frequently causes cytogenetic anomalies. Like cancer cells, immortal cell lines exhibit unlimited proliferative capacity and mostly contain numerical and/or structural chromosomal anomalies [18,19]. In order to gain insight into the biology of immortalized non-cancerous cell lines, we have shown that cytogenetic characterization is critically important. The murine RAW 264.7 monocyte/macrophage-like cell line was established in 1978. These non-cancerous cells present with 40 acrocentric chromosomes, the same modal number as observed in a normal murine karyotype. However, no careful cytogenetic characterization of RAW 264.7 cells has ever been reported. Utilizing conventional (G-banding) and molecular (SKY) cytogenetic techniques, we are the first to demonstrate that the RAW 264.7 cells have a consistent complex translocation, involving chromosomes 1, 3, and 8. Further, we identified three separate clones based on the morphology/banding pattern of chromosome 9. However, the origin and consequences of this translocation as well as the functional differences among the clones are yet to be established. 

Methionine deficiency leads to DNA hypomethylation [11,12], which has a direct effect on chromosomal integrity [10]. Global DNA hypomethylation is implicated in various cancer cells that always exhibit numerical and structural chromosomal instabilities [20,21,22]. Moreover, DNA hypomethylation has also been associated with abnormal chromosomal structures in patients suffering from the DNA hypomethylation-associated disease, called ICF syndrome (Immunodeficiency, Centromeric instability and Facial abnormalities) [10]. The cells of ICF patients display decondensation of pericentromeric heterochromatin, telomere association, multiradial chromosomes, and whole-arm deletions [23]. We hypothesized that methionine deficiency would increase chromosomal instability. Our observations after one month of methionine deficiency for RAW 264.7 cells confirmed our hypothesis. Likewise, an increase in chromosomal aberrations was also observed in the bone marrow cells of mice that were maintained on a methionine-deficient diet for two months.

One explanation for the increased chromosomal aberrations could be that a deficiency of methionine results in a higher DNA break formation. We observed that methionine deficiency enhances DNA break formation in RAW 264.7 cells. Similar to our findings, Pogribny et al. showed that rats fed with a folate/methionine-deficient diet for 9 weeks exhibit a higher level of DNA breaks in the liver cells as compared to their counterparts fed with a normal diet [24]. Increased DNA breaks as a result of dietary methionine deficiency may potentially make DNA more susceptible to nuclear endonucleases. Indeed, it has been shown that dietary folate/methionine deficiency enhances susceptibility to endonuclease-induced DNA breaks [24]. Collectively, these data suggest that long-term methionine deficiency may cause genetic damage to immune cells. Therefore, consideration for adopting this dietary approach as a treatment for cancer patients requires further investigation. 

For growth and development, tumor cells require more methionine than healthy cells [25]. Therefore, controlling tumor growth with dietary methionine deficiency is a promising therapeutic choice. Notably, preclinical studies have demonstrated that dietary methionine deficiency and radiation have synergistic effects in controlling tumors [26]. Perhaps the explanation could be attributed to the fact that methionine deficiency further enhances radiation-induced DSBs, which are primarily responsible for the destruction of tumor cells [27]. However, the effects of methionine deficiency on radiation-induced DNA breaks in non-tumorigenic cells such as monocytes/macrophages have yet to be studied. We observed that methionine deficiency enhances radiation-induced DNA breaks and chromosomal aberrations in RAW 264.7 cells. MRE11 is a DNA damage response protein, and a crucial member of the MRE11-RAD50-NBS1 (MRN) complex [28]. It recognizes and repairs DSBs [28]. The elevated level of MRE11 further confirms that methionine deficiency leads to an increase in radiation-induced DNA DSB formation. Importantly, activation of Poly(ADP-ribose)polymerase 1 (PARP1) also affects DNA methylation. PARP1 also contributes to homologous recombination (HR) system functioning by recruiting critical DNA repair factors, such as MRE11, to sites of DSBs. From a therapeutic point of view, deficiency in genes implicated in HR—including MRE11—confers sensitivity to PARP inhibitors [29]. These data suggest that methionine deficiency can further amplify radiation-induced genetic damage in murine immune cells. 

In conclusion, our studies report the initial cytogenetic characterization of RAW 264.7 cells with discernment of a consistent complex translocation involving chromosomes 1, 3, and 8. This cell line can be further categorized based on three unique clones, dependent on the morphology of chromosome 9. In addition, we demonstrate the significant effects of long-term methionine deficiency for chromosomal damage, independent of radiation treatment, using both in vitro and in vivo models. Methionine deficiency increased the frequency of clone 2, increased chromosomal aberrations, and increased DSB formation in RAW 264.7 cells. Methionine deficiency also enhanced radiation-induced DSB formation and chromosomal aberrations in RAW 264.7 cells. Finally, mice fed with a methionine-deficient diet exhibited a higher number of chromosomal aberrations than those maintained on a diet with sufficient methionine.

Our studies did not address either the functional consequences of the complex translocation, or compare the differences in functional activity among the three clones. Additionally, the underlying mechanisms responsible for an increase in chromosomal aberrations resulting from dietary methionine deficiency—independent of radiation—was not fully addressed. Finally, the role of methionine deficiency in modifying monocyte/macrophage function needs further elucidation. Each of these issues will be of great interest for future studies.

The RAW 264.7 cell line provides an excellent model to study the effects of methionine deficiency on immune cells, specifically monocyte/macrophage cells. As such, our current study provides insight into the previously unexplored role of methionine deficiency in modulating genetic damage prior to and following radiation exposure. These findings suggest that caution is warranted for clinical translation of dietary methionine deficiency in conjunction with cancer radiotherapy.

## Figures and Tables

**Figure 1 ijms-22-02378-f001:**
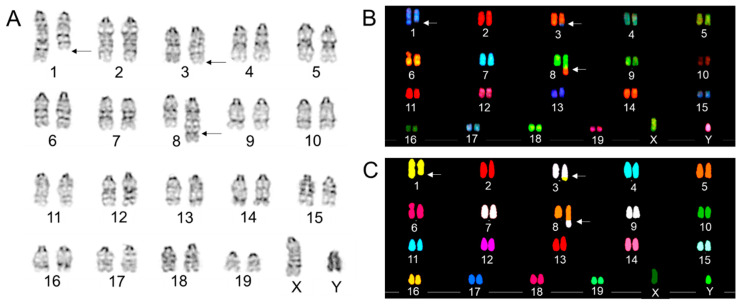
(**A**) A representative G-band karyogram of a RAW 264.7 cell showing 40 chromosomes. Arrows indicate rearrangements, additions, and deletions. Representative (**B**) spectral and (**C**) classified images of a RAW 264.7 cell showing 40 chromosomes. Arrows indicate rearrangements, additions, and deletions.

**Figure 2 ijms-22-02378-f002:**
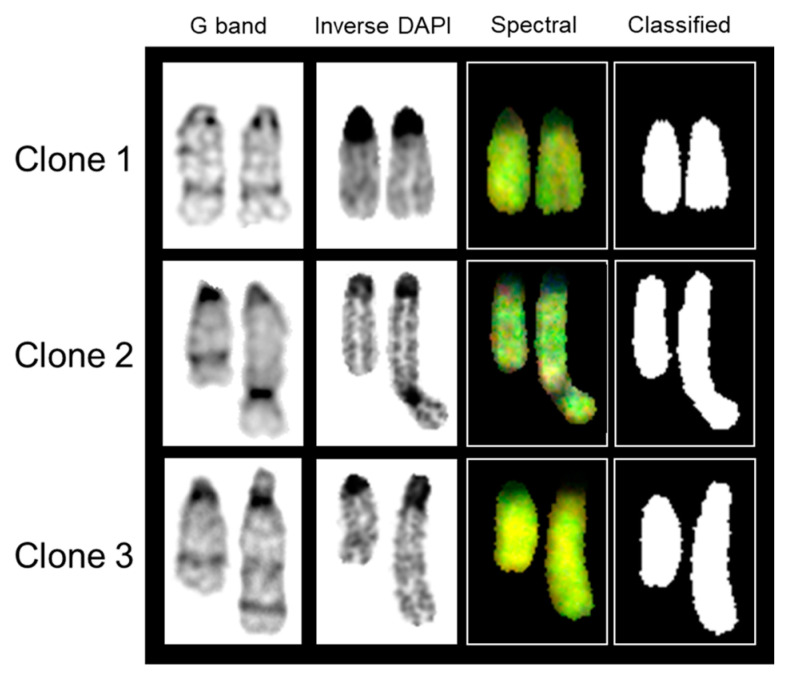
Morphology/banding pattern of chromosome 9 of RAW 264.7 cell gives rise to 3 separate clones.

**Figure 3 ijms-22-02378-f003:**
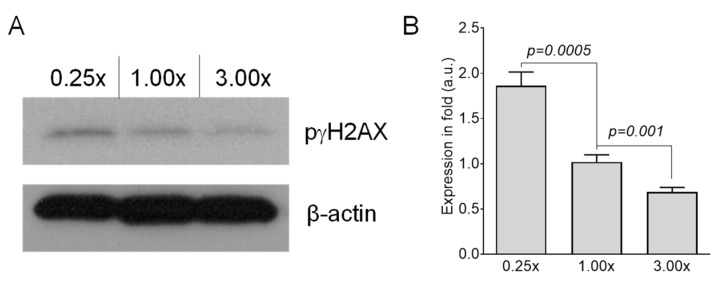
Effects of methionine on genetic damage. (**A**) Representative Western blot analysis and (**B**) quantitation of phosphorylated γH2AX protein, a marker of DNA double-strand breaks, in RAW 264.7 cells grown in either methionine-deficient (0.25×) or adequate (1.00×) or supplemented (3.00×) media for 1 month (*n* = 4). β-actin served as a loading control. *n*, number of independent experiments performed.

**Figure 4 ijms-22-02378-f004:**
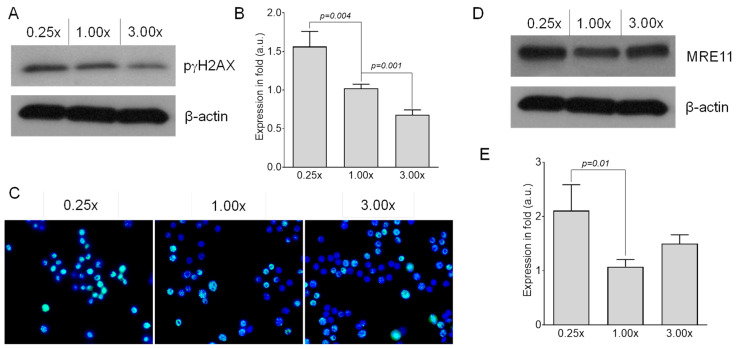
Effects of methionine on radiation-induced genetic damage. (**A**) Representative Western blot analysis and (**B**) quantitation of phosphorylated γH2AX protein, a marker of DNA double-strand breaks, in RAW 264.7 cells grown in either methionine-deficient (0.25×) or adequate (1.00×) or supplemented (3.00×) media for 1 month followed by 2 Gy radiation exposure (*n* = 3). β-actin served as a loading control. *n*, number of independent experiments performed. (**C**) Representative photomicrograph (20× magnification) showing immunofluorescence of γH2AX foci (green) in RAW 264.7 cells grown in either methionine-deficient (0.25×) or adequate (1.00×) or supplemented (3.00×) media for 1 month followed by 2 Gy radiation exposure (*n* = 3). Nuclei were stained with DAPI (blue). (**D**) Representative Western blot analysis and (**E**) quantitation of MRE11 protein in RAW 264.7 cells grown in either methionine-deficient (0.25×) or adequate (1.00×) or supplemented (3.00×) media for 1 month followed by 2 Gy radiation exposure (*n* = 3). β-actin served as a loading control. *n*, number of independent experiments performed. Cells were harvested 4 h after irradiation in these experiments.

**Table 1 ijms-22-02378-t001:** Distribution of aberration frequencies ± standard errors (number of aberrations observed) detected by G-banding after 1-month culture of RAW 264.7 cells in either methionine-adequate media (1.00×) or methionine-deficient media (0.05×). RT, Robertsonian translocation; CSB, chromosomal breaks; a, statistically significant difference in total yield from 1.00× media.

Methionine Content	Metaphase Count	Aneuploidy	RT	CSB	Total Yield	*p* Value
1.00× (0.20 mM)	35	0.03 ± 0.03 (1)	0.03 ± 0.03 (1)	0.00 ± 0.00 (0)	0.06 ± 0.04 (2)	
0.25× (0.05 mM)	36	0.00 ± 0.00 (0)	0.14 ± 0.06 (5)	0.06 ± 0.04 (2)	0.19 ± 0.07 (7)	Total yield *^a^*

**Table 2 ijms-22-02378-t002:** Distribution of aberration frequencies ± standard errors (number of aberrations observed) detected by Spectral Karyotyping (SKY) analysis after 1-month culture of RAW 264.7 cells in either methionine-adequate or methionine-deficient media followed by a single exposure of 2 Gy. Cells were harvested 24 h after radiation. a, statistically significant difference in total yield from 1.00× media.

Methionine Content	Metaphase Count	Aneuploidy	Color Junction	Acentric Fragment	Total Yield	*p* Value
1.00× (0.20 mM)	36	0.39 ± 0.10 (14)	0.81 ± 0.15 (29)	0.83 ± 0.15 (30)	2.03 ± 0.24 (73)	
0.25× (0.05 mM)	38	0.42 ± 0.11 (16)	1.26 ± 0.18 (48)	1.32 ± 0.19 (50)	3.00 ± 0.28 (114)	Total yield ^a^

**Table 3 ijms-22-02378-t003:** Distribution of aberration frequencies ± standard errors (number of aberrations observed) detected by solid Giemsa staining in the bone marrow cells of CBA/J mice fed with either methionine-deficient, MDD (0%), methionine-adequate, MAD (100%), or methionine-supplemented, MSD (300%) diets for 2 months. a, statistically significant difference in total yield from MAD.

Diet	Code	Metaphase Count	Acentric Fragment	RT	Dicentrics	Total Yield	*p* Value
MAD	CAM247	70	0.00 ± 0.00 (0)	0.00 ± 0.00 (0)	0.00 ± 0.00 (0)		
MAD	CAM248	102	0.03 ± 0.02 (3)	0.00 ± 0.00 (0)	0.00 ± 0.00 (0)		
MAD	CAM249	103	0.03 ± 0.02 (3)	0.00 ± 0.00 (0)	0.00 ± 0.00 (0)		
MAD	CAM250	107	0.01 ± 0.01 (1)	0.00 ± 0.00 (0)	0.01 ± 0.01 (1)		
MAD	CAM251	106	0.03 ± 0.02 (3)	0.00 ± 0.00 (0)	0.00 ± 0.00 (0)	0.02 ± 0.01 (11)	
MDD	CAM252	101	0.09 ± 0.03 (9)	0.00 ± 0.00 (0)	0.00 ± 0.00 (0)		
MDD	CAM253	56	0.05 ± 0.03 (3)	0.00 ± 0.00 (0)	0.00 ± 0.00 (0)		
MDD	CAM254	43	0.05 ± 0.03 (2)	0.00 ± 0.00 (0)	0.00 ± 0.00 (0)		
MDD	CAM255	82	0.09 ± 0.03 (7)	0.00 ± 0.00 (0)	0.00 ± 0.00 (0)		
MDD	CAM256	73	0.08 ± 0.03 (6)	0.00 ± 0.00 (0)	0.00 ± 0.00 (0)	0.08 ± 0.01 (27)	Total yield *^a^*
MSD	CAM257	78	0.01 ± 0.01 (1)	0.00 ± 0.00 (0)	0.00 ± 0.00 (0)		
MSD	CAM258	56	0.00 ± 0.00 (0)	0.00 ± 0.00 (0)	0.00 ± 0.00 (0)		
MSD	CAM259	82	0.02 ± 0.02 (2)	0.00 ± 0.00 (0)	0.00 ± 0.00 (0)		
MSD	CAM260	79	0.03 ± 0.02 (2)	0.00 ± 0.00 (0)	0.00 ± 0.00 (0)		
MSD	CAM261	19	0.00 ± 0.00 (0)	0.00 ± 0.00 (0)	0.00 ± 0.00 (0)	0.02 ± 0.01 (5)	

## Data Availability

The data presented in this study are available on request from the corresponding author.

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
