# Peer review of "Dietary Methionine Deficiency Enhances Genetic Instability in Murine Immune Cells"

_ijms, 2021, doi:10.3390/ijms22052378_

Round 1
Reviewer 1 Report
This study by Binz RL et al, provides insight into the previously unexplored role of methionine defi-ciency in modulating genetic damage prior to and following radiation exposure. The manuscript is straightforward, well written, and concise, and has clear results. Definitely deserves to be published and is a valuable contribution to the “International Journal of Molecular Sciences”. Some minor flaws need to be addressed before publication.
Minor points:
[1] “1. Introduction”, Page 2/13:
“Protein expression is tightly regulated by chromosomal DNA. Genetic and epigenetic change in DNA such as changes in DNA methylation can alter protein expression [7,8].”.
By exploiting the relationship between nutrition and gene expression, the epigenetic phenomena related to growth and health processes can be regulated. Please report that this dynamic relationship between nutrition and gene expression throughout the lifetime of an organism has now been recognized as a subfield called “nutritional epigenomics”.
Recommended reference: Ji Y, et al. Nutritional epigenetics with a focus on amino acids: implications for the development and treatment of metabolic syndrome. J Nutr Biochem. 2016 Jan;27:1-8.
[2] “1. Introduction”, Page 3/13:
“Effective repair of DSBs results in cell survival, while inadequate repair leads to chromosomal aberrations or cell death.”.
At that point, please do mention that DSBs are repaired either through homologous recombination (HR) or through non-homologous end joining (NHEJ)—the choice of which is determined by cell cycle phase and chromatin context. NHEJ is the preferred mechanism for repair of DSBs in G1 when a DNA template that could be used for HR is absent.
Recommended reference: Boussios S, et al. Wise Management of Ovarian Cancer: On the Cutting Edge. J Pers Med. 2020;10(2):41.
[3] “4. Discussion”, Page 11/13:
“MRE11 is a DNA damage response protein, and a crucial member of the MRE11-RAD50-NBS1 (MRN) complex [26]. It recognizes and re-pairs DSBs [26]”.
Indeed, MRE11 is an important DNA damage response protein. Activation of PARP1 affects DNA methylation. PARP1 also contributes to homologous recombination (HR) system functioning by recruiting critical DNA repair factors, such as MRE11 to sites of DSBs. From the therapeutic point of view, deficiency in genes implicated in HR - including MRE11 - confers sensitivity to PARP inhibitors.
Recommended reference: Boussios S, et al. Poly (ADP-Ribose) Polymerase Inhibitors: Talazoparib in Ovarian Cancer and Beyond. Drugs R D. 2020;20:55-73.
Author Response
We would like to thank both reviewers for providing thoughtful suggestions, thus improving our revised manuscript.
Reviewer 1
This study by Binz RL et al, provides insight into the previously unexplored role of methionine defi-ciency in modulating genetic damage prior to and following radiation exposure. The manuscript is straightforward, well written, and concise, and has clear results. Definitely deserves to be published and is a valuable contribution to the “International Journal of Molecular Sciences”. Some minor flaws need to be addressed before publication.
Response: We would like to express our gratitude for your expert review and for mentioning that our manuscript deserves publication in the “International Journal of Molecular Sciences.”
Minor points:
[1] “1. Introduction”, Page 2/13:
“Protein expression is tightly regulated by chromosomal DNA. Genetic and epigenetic change in DNA such as changes in DNA methylation can alter protein expression [7,8].”.
By exploiting the relationship between nutrition and gene expression, the epigenetic phenomena related to growth and health processes can be regulated. Please report that this dynamic relationship between nutrition and gene expression throughout the lifetime of an organism has now been recognized as a subfield called “nutritional epigenomics”.
Recommended reference: Ji Y, et al. Nutritional epigenetics with a focus on amino acids: implications for the development and treatment of metabolic syndrome. J Nutr Biochem. 2016 Jan;27:1-8.
Response: We found this suggestion to be helpful. Consequently, we have mentioned the field of “nutritional epigenomics” and incorporated the corresponding reference into our revised manuscript.
[2] “1. Introduction”, Page 3/13:
“Effective repair of DSBs results in cell survival, while inadequate repair leads to chromosomal aberrations or cell death.”
At that point, please do mention that DSBs are repaired either through homologous recombination (HR) or through non-homologous end joining (NHEJ)—the choice of which is determined by cell cycle phase and chromatin context. NHEJ is the preferred mechanism for repair of DSBs in G1 when a DNA template that could be used for HR is absent.
Recommended reference: Boussios S, et al. Wise Management of Ovarian Cancer: On the Cutting Edge. J Pers Med. 2020;10(2):41.
Response: We would like to thank reviewer for this beneficial comment. We have incorporated the suggested verbiage as well as the corresponding reference into our revised manuscript.
[3] “4. Discussion”, Page 11/13:
“MRE11 is a DNA damage response protein, and a crucial member of the MRE11-RAD50-NBS1 (MRN) complex [26]. It recognizes and re-pairs DSBs [26]”.
Indeed, MRE11 is an important DNA damage response protein. Activation of PARP1 affects DNA methylation. PARP1 also contributes to homologous recombination (HR) system functioning by recruiting critical DNA repair factors, such as MRE11 to sites of DSBs. From the therapeutic point of view, deficiency in genes implicated in HR - including MRE11 - confers sensitivity to PARP inhibitors.
Recommended reference: Boussios S, et al. Poly (ADP-Ribose) Polymerase Inhibitors: Talazoparib in Ovarian Cancer and Beyond. Drugs R D. 2020;20:55-73.
Response: We would like to thank reviewer for this insight. We have incorporated the suggested verbiage along with the corresponding reference into our revised manuscript.
Reviewer 2 Report
Binz et al. use two differential model systems in vitro (RAW264.7) and in vivo (CBA/J mice) to study the impact of methionine deficiency on non-tumorous tissue. In addition this paper shows how such methionine deficiency can render the karyotype vulnerable when cell are irradiated. For the first time and in absence of any treatment RAW264.7 karyotype is extensively characterized, which led to the observation of chromosomal heterogeneity. Three clones that only differ from chromosome 9 are characterized. All these clones have baseline aberrations (translocations) on chromosome 1, 3 and 8. Methionine deficiency increases aberrations such as Robertsonian translocations and breaks and also shifts the all-over distributions of the initial 3clones towards aberrations of chromosome 9. The authors link these aberrations to increased DNA damage (as quantified by yH2AX). After irradiation the karyotype of methionine-deprived cells is more affected then these of cells grown under “normal” methionine levels, which again is accompanied by higher levels of DNA damage markers (yH2AX and MRE11).
In a final in vivo analysis mouse is fed with adequate, low and high amounts of methionine. Karyotype analysis of bone marrow cells reveal that low methionine amounts lead to more karyotype aberrations, than normal or high methionine levels.
The here presented study provides an interesting new aspect of how tumour- specific diets have to be handled with care, as we need to consider that also non-tumorous cell types (in this case; monocytes/macrophages) are suffering from absence of essential nutrients.
Major concerns:
The main message of the paper gets a bit lost as many different aspects are mentioned in the manuscript (mainly the introduction), but finally not further elucidated (such as methylation states after methionine deprivation).
Another example is the fact that monocytes/macrophages are important for immune defence and can infiltrate the tumour (as stated in the manuscript), yet the here used cells are already derived from tumorous tissue. This aspect is very well reflected by the fact that the karyotype shows stable (involving Chr. 1,3 and 8) and unstable aneuploidies (Chr.9).
This evokes the question whether the here chosen model was adequate to answer the question on how methionine deficiency influences “normal (non-cancerous)” cells.
Minor remarks:
- Table 1 should have better distinguished columns
- it does not become clear from the manuscript whether the mice (Table 3) was irradiated, yet in the Material and Methods it is stated
Further remarks
Methionine deficiency can lead to hypo-methylation, which in turn can change the status of hetero- and euchromatin. Are such differences visible with the G-banding?
Author Response
We would like to thank both reviewers for providing thoughtful suggestions, thus improving our revised manuscript.
Reviewer 2
Binz et al. use two differential model systems in vitro (RAW264.7) and in vivo (CBA/J mice) to study the impact of methionine deficiency on non-tumorous tissue. In addition this paper shows how such methionine deficiency can render the karyotype vulnerable when cell are irradiated. For the first time and in absence of any treatment RAW264.7 karyotype is extensively characterized, which led to the observation of chromosomal heterogeneity. Three clones that only differ from chromosome 9 are characterized. All these clones have baseline aberrations (translocations) on chromosome 1, 3 and 8. Methionine deficiency increases aberrations such as Robertsonian translocations and breaks and also shifts the all-over distributions of the initial 3clones towards aberrations of chromosome 9. The authors link these aberrations to increased DNA damage (as quantified by yH2AX). After irradiation the karyotype of methionine-deprived cells is more affected then these of cells grown under “normal” methionine levels, which again is accompanied by higher levels of DNA damage markers (yH2AX and MRE11).
In a final in vivo analysis mouse is fed with adequate, low and high amounts of methionine. Karyotype analysis of bone marrow cells reveal that low methionine amounts lead to more karyotype aberrations, than normal or high methionine levels.
The here presented study provides an interesting new aspect of how tumour- specific diets have to be handled with care, as we need to consider that also non-tumorous cell types (in this case; monocytes/macrophages) are suffering from absence of essential nutrients.
Major concerns:
The main message of the paper gets a bit lost as many different aspects are mentioned in the manuscript (mainly the introduction), but finally not further elucidated (such as methylation states after methionine deprivation).
Response: We are thankful to the reviewer for this comment. Previous studies, including ours, demonstrated that altered methionine dietary intake subsequently results in altered methylation patterns (PMID: 29939201; PMID: 28904640; PMID: 22872676). We agree with the reviewer that delineating how methionine deprivation shapes the bone marrow epigenome is critical in understanding the mechanisms of toxicity. Therefore, we are currently performing a large-scale study with the assessment of DNA methylation patterns in various populations of bone marrow cells (i.e., hematopoietic stem cells, progenitor cells, etc) paralleled with functional analysis. Because of the magnitude of the ongoing study and in an effort not to dilute the message of this study, the DNA methylation data will be published in a separate article.
Another example is the fact that monocytes/macrophages are important for immune defence and can infiltrate the tumour (as stated in the manuscript), yet the here used cells are already derived from tumorous tissue. This aspect is very well reflected by the fact that the karyotype shows stable (involving Chr. 1,3 and 8) and unstable aneuploidies (Chr.9).
This evokes the question whether the here chosen model was adequate to answer the question on how methionine deficiency influences “normal (non-cancerous)” cells.
Response: We acknowledge the validity of the reviewer’s comment. However, we have observed an elevated level of chromosomal damage in bone marrow stem and progenitor cells, which were obtained from un-irradiated mice following 2 months of methionine deprivation, suggesting that methionine deficiency can lead to genomic instability in normal cells.
Minor remarks:
- Table 1 should have better distinguished columns
Response: We agree and appreciate this helpful remark. Consequently, Table 1 has been reformatted for easier discernment.
- it does not become clear from the manuscript whether the mice (Table 3) was irradiated, yet in the Material and Methods it is stated
Response: We are appreciative for this astute observation, and apologize for neglecting this detail in our submitted manuscript. The mice were not exposed to radiation. This information has been corrected in Materials and Methods.
Further remarks
Methionine deficiency can lead to hypo-methylation, which in turn can change the status of hetero- and euchromatin. Are such differences visible with the G-banding?
Response: The reviewer indeed makes a valid point. Unfortunately, it is not prudent to identify subtle changes based solely on G-band analysis, as the identification of subtle changes to the banding pattern are dependent upon the band level of resolution for each chromosome.
Round 2
Reviewer 2 Report
Thank you for responding to all my concerns.
Please just correct Table 1, which still has no clearly divided columns.
Best regards.